# The orchestrating role of platform enterprises in digital inclusion: A network game perspective from industrial clusters

**Dingteng Wang** [1,2]*, **Ningning Li**[1◉], **Guoan Liang**[1,3◉], **Shengming Li**[1]

1 School of Business, Guangxi City Vocational University, Chongzuo, Guangxi, China, 2 School of Business, Guangxi University, Nanning, Guangxi, China, 3 The School of Management, Guangxi University for Nationalities, Guangxi, China

◉ These authors also contributed equally to this work.
* dingtengwang@163.com

## Abstract

The digital transformation of industrial clusters, while enhancing productivity, has exacerbated the corporate digital divide, particularly for small and medium-sized enterprises (SMEs). Digital inclusion, which aims to bridge this divide, exhibits public-good attributes and positive externalities, yet its co-construction is often hindered by the "Olson's dilemma" arising from the conflict between individual and collective rationality. Platform enterprises, occupying the "bridging point" of structural holes within clusters, are posited to act as opinion leaders capable of orchestrating ecosystem co-creation. Through a theoretical simulation approach, there employs a "small-world + opinion leader" complex network framework to model the evolutionary dynamics of digital inclusion ecosystem co-construction. Integrating a public goods game model with a Fermi learning algorithm enhanced by a random surfer mechanism, we simulate strategic interactions among cluster firms under controlled parameter conditions. Monte Carlo simulations reveal a distinct threshold effect under simulated conditions: cooperation emerges sustainably only when the investment return coefficient surpasses the critical value defined by the number of game participants. Furthermore, the platform's connection ratio and strategic commitment emerge as pivotal factors; maintaining an investment strategy significantly promotes cooperation, whereas free-riding by the platform exerts a strong negative demonstration effect. While traditional platform governance tools like rewards and punishments can mitigate cooperation decay, they cannot reverse collective free-riding trends in our model. Crucially, a binding collective agreement, featuring pre-commitment of costs and redistribution of unused funds, emerges as the most effective mechanism for resolving the cooperation dilemma within the simulation. These theoretically derived findings offer insights for policymakers, platform enterprises, and SMEs in fostering collaborative and inclusive digital ecosystems within industrial clusters, though empirical validation remains necessary.

**Data availability statement:** The Data has been uploaded to the Open Science Framework (OSF) and is available at the following link: https://osf.io/axtf9/overview?view_only=f39e90a6a71e-40cd811aa5469a4aa9f6.

**Funding:** This research was supported by The Guangxi Science and Technology Think Tank Key Project (Grant No. K-88) funded computational resource leasing.The Guangxi Basic Ability Improvement Project for Young and Middle-aged Teachers in Universities (Grant No. 2024KY1937) covered publication-related expenses.The Guangxi Vocational Education Teaching Reform Research Project (Grant No. GXGZJG2024A004) supported costs associated with literature retrieval, access, and research-related travel.The funders had no role in study design, data collection and analysis, decision to publish, or preparation of the manuscript.

**Competing interests:** The authors have declared that no competing interests exist.

## Introduction

With the deepening penetration of digital technologies into industrial clusters, the issue of the digital divide among enterprises is becoming increasingly pronounced. From the perspective of factor allocation, the digital economy leverages big data to enhance information transparency, which helps reduce manufacturing production costs, optimise resource allocation, and stimulate innovation, thereby improving enterprise production efficiency and promoting the structural upgrading of the manufacturing sector [1]. Research based on a mediating effect model indicates that digital technology significantly promotes industrial structural upgrading, with the level of independent innovation and the advancement of human capital structure playing partial mediating roles in this process [2]. Innovation environment elements such as talent agglomeration and financial development also have moderating effects in this context [3]. However, digital technology does not permeate evenly; its adoption within industrial clusters exhibits structural disparities, leading to a digital divide in the acquisition and application capabilities of information and communication technologies among different entities.The corporate digital divide refers to the imbalance in digitalisation levels arising from disparities among enterprises in accessing, using, and generating information, technology, and resources [4,5]. This divide not only reinforces winner-takes-all and path dependence phenomena, allowing digitally advanced firms to continuously expand their competitive advantage, but also makes it difficult for latecomer firms to catch up effectively [6]. Specifically, differences in early-stage information technology investment levels have led to significant divergence in the adoption of internet technologies among firms [7]. Concurrently, the gap in human capital is another critical factor constituting the corporate digital divide—managers and talents with professional skills are key to the successful introduction and application of information technology [8,5]. Against this backdrop, some enterprises struggle to fully share the developmental dividends brought by digitalisation, resulting in the digital economy's promotion of manufacturing innovation exhibiting characteristics of a digital divide rather than genuine inclusive benefits.

In this context, digital inclusion has emerged as a critical pathway to bridge the digital divide. Effectively resolving the digital divide relies on various actors implementing ecosystem coordination strategies and engaging in value co-creation with ecological partners [9], jointly advancing the realisation of digital inclusion [10]. This also urgently requires academia to deeply explore new paradigms for the synergistic development and symbiosis of stakeholders within the digital context [11]. On 22 September 2024, the United Nations adopted the Global Digital Compact, explicitly aiming to "Close all digital divides and deliver an inclusive digital economy" [12]. Digital inclusion has become a crucial approach for societies and businesses to address the digital divide [13], reflecting the practical direction for value co-creation and cross-boundary integration between firms and their stakeholders in digital scenarios, thus receiving significant global attention.From a practical standpoint, actions promoting digital inclusion in remote areas of developed countries and in developing nations primarily revolve around two categories of measures: firstly, the construction of information and communication technology (ICT)-related infrastructure [14,15],

and secondly, the conduction of ICT-related knowledge transfer and capacity building [16,4]. Essentially, digital inclusion involves core organisations occupying bridging positions in the structural holes of digital transformation. By leveraging their intermediary role, they facilitate cross-border integration and comprehensive digital transformation for actors in weak relational states. It is not only a concrete manifestation of responsible innovation but also a crucial systematic pathway for bridging the digital divide and achieving multi-party value co-creation [17].

However, a fundamental tension exists between the diverse interests of cluster firms and the public-good nature of the digital inclusion ecosystem. Digital transformation serves as a critical strategic choice for industrial clusters to break free from path dependency and achieve an upward shift in their global value chain positioning [18]. While the digital transformation of cluster firms relies on the synergistic co-construction of the entire industrial cluster ecosystem, a pronounced digital divide persists between Small and Medium-sized Enterprises (SMEs)—the core constituents of clusters—and their larger counterparts [19]. This widening divide not only forces SMEs, which constitute the long-tail segment, to bear high digital investment costs while suffering from low production efficiency [20], but also traps them in systemic exclusion from cluster networks and a low-end lock-in. Consequently, it constrains high-level innovation within the digital ecosystem and hampers the sustainable development of the industrial cluster itself [21].New digital infrastructure plays a pivotal role in driving both the digital transformation and high-quality development of cluster firms [22]. Nonetheless, industrial clusters exhibit significant ecological diversity and demand heterogeneity. Firms, often driven by a scale-oriented growth logic, typically pursue rapid expansion [23]. This prevailing strategy renders digital infrastructure development models—which possess public-good attributes but fail to account for firm heterogeneity—difficult to sustain [23]. As rational economic actors, firms generally possess a free-rider motivation, tending to externalise the construction costs of inclusive public goods. This allows them to avoid investment risks while simultaneously capturing the positive external benefits generated by the digital ecosystem. This strategic choice originates from a fundamental structural conflict between individual and collective rationality, exemplifying the classic Olson's Dilemma of collective action.

Platform enterprises, serving as the core of industrial clusters, occupy the bridging point of structural holes in digital transformation and hold a position of opinion leadership. Structural holes arise from the absence of direct relationships between fragmented markets or actors [24]. The establishment of weak ties and data exchange between supply and demand parties on platforms is key to forming this bridging point for platform enterprises [25]. Those occupying such positions can guide the effective flow of resources, break down information silos, promote collaboration among stakeholders, and construct network structures conducive to innovation [26]. As leaders of the industrial platform, platform enterprises influence the decision-making and behavioural logic of participants through strategic adjustments [27], thereby gaining their trust and attracting them to join in co-creation [28]. For SMEs, relying on platform enterprises for empowerment has become a vital pathway to pursue digital catch-up and business transformation [29]. Therefore, clarifying the following research questions is essential for addressing cooperation dilemmas within industrial clusters, bridging the digital divide, and fostering a digitally inclusive society:

RQ1: What are the driving mechanisms and influencing factors for building a digital inclusion ecosystem within industrial clusters?

RQ2: To what extent does the platform enterprise, as an opinion leader, exert an exemplary effect?

RQ3: How can its positive influences be maximised while mitigating negative effects in practice?

RQ4: How can top-level design be utilised to prevent the emergence of cooperation dilemmas?

Although a limited number of studies have acknowledged the externality and public-good nature of digital inclusion [30], as well as the tension between high investment and broad accessibility in the construction of digital public goods [31], existing analytical frameworks remain insufficient. We contend that cluster firms' investment behaviours in the digital inclusion ecosystem not only exhibit the basic characteristics of a public goods game but are also embedded within a complex structure of network interactions, wherein platform enterprises play an indispensable catalytic role.The contributions of this paper are threefold. First, it constructs an integrated public goods game analysis framework incorporating "small-world

networks + opinion leaders", examining the dynamics of public goods provision in digital inclusion ecosystems from a complex network perspective. Second, it innovatively incorporates the random surfer model into the Fermi learning algorithm, thereby refining strategy-updating rules and more accurately reflecting the complex cross-domain learning behaviours of firms in industrial clusters. Third, it provides an in-depth exploration of how platform enterprises—fulfilling dual roles as both market participants and quasi-public managers—make strategic choices and exert influence, and how such actions shape system evolution. This offers new theoretical insights into how platform enterprises can foster synergistic development within industrial clusters.

As a theoretical simulation study, this work constructs a theoretical analytical framework grounded in game theory. To do so, it employs small-world networks to simulate the structure of industrial clusters, leverages opinion leadership theory to explain the guiding role of platform enterprises, and utilizes public goods games to model the cooperation dilemma inherent in the co-construction of a digital inclusion ecosystem. Together, these theoretical perspectives form a multi-layered explanatory system. The remainder of this paper is structured as follows. In the Theoretical analysis section, we examine the small-world characteristics of industrial clusters, the opinion leadership of platform enterprises, and the public-good attributes of digital inclusion. In the Model construction section, we detail the integration of a small-world network with an opinion leader into a public goods game, enhanced by a Fermi learning algorithm incorporating a random surfer mechanism. Simulation and analysis then explore the impacts of key parameters and the platform's connection ratio on cooperation dynamics under controlled conditions. Finally, further research investigates the effects of platform strategies and governance mechanisms—such as rewards, punishments, and collective agreements—on system evolution within the simulation environment.

## Theoretical analysis

### Small-world network characteristics of industrial clusters

Industrial clusters are defined as geographically concentrated agglomerations of firms and institutions—such as suppliers, manufacturers, financial entities, and research organisations—interconnected through competitive and cooperative relationships, highlighting their pivotal role in global competition [32]. This theoretical construct not only reveals the spatial organisational attributes of industrial clusters but also underscores their strategic value as carriers of regional competitiveness. From a vertical perspective, clusters exhibit gradient differences in the depth of industrial chain integration, the density of knowledge networks, and the intensity of innovation synergy. Horizontally, industrial clusters have increasingly transcended traditional industrial boundaries, evolving into regional innovation ecosystems characterised by multi-industry convergence and multi-actor symbiosis. Such organisational forms, with their multi-dimensional nested structures, have become an important indicator for measuring regional economic development levels [33]. In essence, they represent the geographical concentration of interrelated industries connected by factors such as knowledge and demand [32]. With the deep penetration of digital technologies, the spatial organisation logic of industrial clusters is being reconfigured, making their digital transformation a key topic of interest for both academia and industry [34]. On one hand, digital platforms propel clusters toward a more networked development pattern; on the other, the flow and sharing of data elements accelerate value co-creation and knowledge diffusion. As a vital organisational form supporting the development of the real economy [33], the transformation pathways and mechanisms of industrial clusters in the digital context have been widely studied [35].

Small-world networks are primarily characterised by a high clustering coefficient and a short average path length. Their theoretical foundation can be traced back to Milgram's "six degrees of separation" experiment, which first revealed the high connectivity of human social networks [36]. Building on this, Watts and Strogatz proposed the small-world network theory through mathematical modelling, indicating that real-world networks are neither completely regular nor entirely random but exhibit a topological structure combining high clustering with short path lengths—meaning the presence of numerous tightly-knit subgroups while any two nodes can be connected through only a few steps [37]. This structure,

which integrates local clustering with global connectivity, provides a powerful tool for describing real-world social networks. From social and biological neural networks to internet topologies and industrial cluster networks, small-world properties are universally present [38], offering an effective quantitative framework for studying real-world social relations and inter-active behaviours.

Specifically, industrial clusters—as complex social networks composed of multiple elements, actors, and connections—demonstrate typical small-world characteristics. Structurally, entities within clusters, such as suppliers, manufacturers, and financial institutions, form interlinked structures with high clustering coefficients through complex interactions [39]. Despite the large number of firms within a cluster, geographical proximity and cross-sector collaboration enable information and resources to propagate rapidly via short average path lengths [40]. Substantial empirical research shows that small-world properties are prevalent across diverse contexts, including creative industries [41], financial clusters [42], and electric vehicle networks [43], with some studies directly employing small-world networks to represent cluster structures in analysis [44]. Therefore, to more accurately capture the game interactions among firms within industrial clusters, this paper adopts the small-world network model to simulate their internal interactive behaviours.

## The opinion leadership status of platform enterprises

In the digital economy, platform enterprises serve as pivotal intermediary organisations that construct open ecosystems to connect multiple stakeholders, reduce interaction costs, optimise resource allocation, and foster innovation-driven value co-creation. Their organisational form can be traced back to the "platform organisation" proposed by Ciborra —a structure capable of flexibly combining resources, rules, and architectures to respond to emerging business opportunities [45]. In the course of their development, platform enterprises often face a tension between "differentiation and commonality": they must address the personalised needs of ecosystem participants to enhance value perception [46], while also building common capabilities to support scalable expansion [47].

An opinion leader is an actor possessing specialised knowledge, informational advantages, or social influence within a specific domain, capable of shaping group decisions through dissemination networks. During information diffusion, certain agents can exert influence over the opinions, decisions, and actions of the majority of others. These agents are referred to as opinion leaders [48]. A central position within the network is a key characteristic of opinion leaders [49]. The "con-centric circles" structure suggests that the influence of opinion leaders diffuses from near to far and from dense to sparse; their similarity to the "social template" is positively correlated with the degree of user agglomeration [50]. By virtue of their extensive connections and interactive capabilities, opinion leaders function as information hubs and centres of public opin-ion, thereby acquiring informational advantages and social influence—forms of network power [51,52].

Platform enterprises exhibit distinctive opinion leader qualities by occupying the "bridging point" of structural holes within industrial clusters. In management practice, leaders play a critical role in shaping group behaviour. Regard-less of the degree of power concentration, leaders can promote the internalisation of behavioural norms through a "demonstration-imitation" mechanism, thereby influencing outcomes in public goods games [53]. As cluster "leaders," platform enterprises can transcend boundaries and facilitate resource optimisation [54]. Within the complex social network of an industrial cluster, the core enterprise—often termed the "chain leader"—exerts a profound influence on the techno-logical development and operational activities of other firms in the cluster through its information channels and resource integration capabilities [55]. Chain leaders not only possess market and technological advantages but also generate significant externalities; they play a key role in coordinating member relationships, constructing innovation mechanisms, and promoting cluster upgrading [56]. As platform-type chain leaders, their quasi-public nature further enhances their capacity to advance industrial cluster development through the provision of public goods [57]. Taking Alibaba as an exam-ple, its digital ecosystem effectively expands cluster scale, accelerates market response, and enhances the efficiency of resource integration and knowledge spillover, significantly strengthening the collaborative innovation capability and competitive advantage of the industrial cluster [58]. Thus, platform enterprises are not merely technological intermediaries

 

or organisational vehicles; they also function as opinion leaders guiding the synergistic development and digital transformation of industrial clusters. To systematically examine their influence, we employs a network science approach, modeling the platform as an opinion-leader node within a small-world network to analyze its orchestrating role in the digital inclusion ecosystem.

## Public attributes and externalities of digital inclusion

The digital inclusion ecosystem exhibits significant positive externalities. This economic attribute implies that the enhancement of digital capabilities by individuals or firms can generate social welfare spillovers that extend beyond their own benefits. According to endogenous growth theory, such externalities are primarily realised through the following mechanisms: first, the knowledge spillover effect, where improvements in corporate digital literacy diffuse through industrial cluster networks, facilitating technology transfer and innovation synergy [59]; second, the synergistic effect of value co-creation and network effects, where co-creation mechanisms in the industrial domain are constructed through the coupling of knowledge between digital platforms and ecosystem participants, along with modularised and systematic rules [60], so that digital improvements in individual nodes can enhance the value of the entire network, creating a "1 + 1 > 2" synergy [61]; and third, human capital accumulation, where the outcomes of digital skills training are disseminated across firms through labour mobility [62]. Such externalities are reflected not only in economic benefits such as improvements in total factor productivity, but also in social benefits such as bridging the digital divide and optimising employment structures.

The public-good nature of the digital inclusion ecosystem stems from the non-excludability and non-rivalry inherent in digital technologies. The concept is defined by the National Digital Inclusion Alliance as "a combination of programmes and policies that meet the unique and diverse needs of geographic clusters," emphasising the need for coordinated efforts by multiple stakeholders—including local service entities, social advocates, and cluster leaders—to systematically address the digital divide [63]. On one hand, the ecosystem aims to build an inclusive digital access system, ensuring the realisation of non-excludability through institutional design. For instance, facilities such as public Wi-Fi networks, open data interfaces, and open-source software systems not only lower the barriers to digital entry but also provide foundational support for the digital evolution of the entire ecosystem [64]. On the other hand, digital resources are characterised by typical non-rivalry [65], meaning that usage by one individual incurs no marginal cost and does not diminish the utility for other users. A case in point is the Apache server, which supports over 40% of global websites; an increase in its user base not only does not impair access quality but enhances overall value through network effects.

However, building such an ecosystem imposes higher entry thresholds regarding industry mechanisms, thereby placing greater emphasis on ecological win–win outcomes and altruistic thinking [66]. It requires a focus on integrating digital technology with business expertise [67]; a platform construction path that relies solely on resource advantages for rapid expansion lacks ecological sustainability [68]. Consequently, the rights to such an ecosystem cannot be fully appropriated by the platform enterprise alone [69]. As rational actors, firms generally exhibit free-rider incentives, seeking to externalise the construction costs of inclusive public goods while fully capturing the positive external benefits generated by the digital ecosystem. This strategic choice fundamentally originates from the structural conflict between individual and collective rationality—a classic collective action dilemma.

## Model construction

### Network construction and opinion leader identification

Consider a set of $n$ cluster firms forming an industrial cluster network $I_0 = (P_0, D_0)$, where $P_0 = \{p_1, p_2, ..., p_n\}$ denotes the set of nodes representing the $n$ cluster firms, and $D_0 = \{d_{11}, d_{12}, ..., d_{nn}\}$ denotes the set of degrees. A neighbour is defined as a firm with which a given firm has business interactions. If firms $p_i (p_i \in D_0)$ and $p_j (p_j \in D_0)$ engage in business exchanges—meaning they are neighbours—then $d_{ij} = d_{ji} = 1$; otherwise, $d_{ij} = d_{ji} = 0$. The set of neighbours for a firm node $p_i$

is $P_i = \{j \neq i | d_{ji} = 1\}$, and the total number of neighbours for $p_i$ is $d_i = \sum_{i=1}^{n} d_{ij}$. Business interactions between neighbouring firms are realised through the multi-dimensional coupling of resource flows, capital flows, and information flows. This coupling relationship determines the adaptability of the digital inclusion ecosystem; firms with business ties exhibit systemic compatibility, whereas those without such links demonstrate systemic heterogeneity. This neighbourhood effect results in the investment dividends of the digital inclusion ecosystem possessing characteristics of localised spillovers, meaning only adjacent nodes in the network topology can fully absorb the positive externalities generated by system construction.

A platform enterprise is embedded within this network structure, connecting upstream and downstream industries and establishing business collaboration relationships with cluster firms. By virtue of its opinion leadership, it guides the industrial cluster. From the perspective of network topology, this platform enterprise acts as a super-node, forming close adjacency relationships with the cluster firms. This yields a new network $I = (D, S)$, where the platform enterprise's network node is denoted as $p_{n+1}$. The new node set is $P = \{p_1, p_2, ..., p_n, p_{n+1}\}$, and the new degrees set is $\{d_{11}, d_{12}, ..., d_{n+1n+1}\}$.

Methods for identifying opinion leaders generally fall into two categories: the first is scoring rule methods based on feature information extraction, such as user leadership [70], user activity [71], and topic relevance [72]; the second is node importance algorithms based on the topology of the social network graph, such as PageRank [73], LeaderRank [74], and HITS [75]. These methods are matched and optimised according to specific research problems. For the digital inclusion ecosystem co-construction problem studied in this paper, which involves constructing an industrial cluster network graph and studying the game interactions within it, node importance algorithms based on social graph topology are more suitable. The PageRank algorithm, a type of eigenvector centrality, considers both the number of connections a node has and their importance, offering relatively high reliability. It can be applied independently to mine opinion leaders [76]. Therefore, this paper employs the PageRank algorithm to rank the eigenvector centrality of all nodes in the industrial cluster network $p_i$ and to assess the opinion leadership status of the platform enterprise. The specific calculation is as follows:

$$PR(i) = (1-\gamma) + \gamma \sum_{j \in P_i} \frac{PR(j)}{d_j} + \gamma E(i)$$

(1)

In this formula, $PR(i)$ represents the eigenvector centrality of firm $p_i$, indicating the importance of node $p_i$ within the network. A higher $PR$ value denotes greater importance and a stronger opinion leadership position for the node. The PageRank algorithm follows the random surfer model: a user either follows a link from the current page with probability $\gamma$, or, due to shifting interests, jumps randomly to another page. This retrieval mechanism more closely mirrors the behavioural model of real firms selecting reference objects. The damping factor $\gamma (0 \leq \gamma \leq 1)$ represents the probability of selecting a neighbour as a reference object; setting $\gamma = 0.85$ is considered appropriate [77]. The term $\frac{PR(j)}{d_j}$ can be understood as the $PR$ value that firm $p_i$ receives from its neighbour $p_j$. $E(i)$ is a decay factor included to prevent the transmitted $PR$ value from being trapped within clusters of nodes lacking outbound links, which would cause $PR$ accumulation. In the PageRank algorithm, each node's initial $PR$ value is 1. The $PR$ values for all nodes are calculated recursively using the formula until they stabilise.

The final network generation process proceeds as follows: First, generate an extensible two-dimensional plane where each cluster firm is placed as a node, forming a ring. Each node connects to its nearest $\frac{\sigma}{2}$ other nodes. At this stage, node $p_i$ has a degree $d_i = \sigma$, where $\sigma$ is the average out degree of the nodes. Second, for each node, with probability $q$, break an edge and reconnect it—that is, randomly disconnect the current node from one connected node and randomly select another node to establish a new connection with the current node. Self-connections and duplicate connections are not permitted. Third, after each node in the original lattice network has been considered once for edge rewiring with probability $q$, a small-world network of cluster firms is obtained. At this point, the degree $d_i$ of node $p_i$ may differ from $\sigma$, and all edges in the network are bidirectional. Finally, create the platform enterprise node and connect it to nodes within the network,

resulting in the "small-world + opinion leader" complex network model. The opinion leader status of the platform enterprise node is then verified based on its *PR* value.

## Game matrix

Based on the public-good attributes and positive externalities of the digital inclusion ecosystem, this study employs a public goods game model as the analytical framework. The core assumptions of the model include that game participants can obtain non-excludable benefits spilled over from the strategic choices of their neighbours, and that they face a binary strategy of either investing ($\lambda_i = 1$) or not investing ($\lambda_i = 0$). When a firm adopts the investment strategy, it signifies its participation in the co-construction of the digital inclusion ecosystem and bears the corresponding construction costs. When a firm chooses the non-investment strategy, it acts as a "free-rider", avoiding construction costs. The strategy set for all participating cluster firms is $\lambda = \{\lambda_1, \lambda_2, \ldots, \lambda_{n+1}\}$.

Platform enterprises play a dual role in the digital inclusion ecosystem, acting as both public agents and economic agents. The public agent attribute requires the platform enterprise to aim for the maximisation of the interests of the attached industrial cluster; as a market-oriented entity, the economic agent attribute drives it to pursue the maximisation of its own benefits. Consequently, platform enterprises must balance between empowering the industrial cluster and pursuing their own development.

Each firm participates in the "small-world + opinion leader" game network as a game participant. Considering a finite population, the total number of game participants is $n+1$. However, within the small-world network, the number of neighbours varies among firms, leading to differences in their respective game matrices. The number of game participants from the perspective of firm $p_i$ is $1+d_i$, i.e., itself plus its neighbours. The investment cost for each participant is $c(c > 0)$, and the return coefficient on investment is $r(r > 1)$. Due to operational separation among cluster firms, only neighbours can share the investment returns from a firm that chooses to invest; firms further away in the network cannot benefit from spillovers from non-neighbours. Therefore, the expected investment return for a firm $i(i \in N)$ that adopts the investment strategy is $\frac{rc}{1+d_i} - c$, and the neighbours of $p_i$ will receive a return of $\frac{rc}{1+d_i}$ from $p_i$. A firm $p_i$ that chooses not to invest will not provide any benefits to its neighbours. Denote the investment strategy of firm $p_i$ as $\lambda_i = 1$ the non-investment strategy is denoted as $\lambda_i = 0$. The actual payoff $\pi_i$ obtained by firm $p_i$ after one round of the game is as follows:

$$\pi_i = \lambda_i \left( \frac{rc}{1+d_i} - c \right) + \sum_{j=1}^{n+1} \frac{d_{ji} \lambda_j rc}{1+d_j}$$

(2)

In this equation, $\lambda_i(\frac{rc}{1+d_i} - c)$ represents that if firm $p_i$ adopts the investment strategy, it incurs a cost $c$ and gains an investment return of $\frac{rc}{1+d_i}$ from its own strategy. If firm $p_i$ adopts the non-investment strategy, it neither incurs any cost nor receives any investment return from its own strategy. The term $d_{ji} \lambda_j \sum_{j=1}^{n+1} \frac{rc}{1+d_j}$ represents the spillover benefits that firm $p_i$ obtains from its neighbours who have adopted the investment strategy.

Within this model, the platform enterprise's objective of profit maximisation exhibits dual dependency. On the one hand, its profit level is closely related to the strategic choices of its neighbouring nodes; specifically, when all neighbouring nodes adopt the investment strategy, a positive externality effect is formed. On the other hand, profit maximisation also depends on the density of business connections between the platform enterprise and the cluster firms; that is, establishing an extensive network of business interfaces with cluster firms can significantly enhance the platform enterprise's profit level.

From the perspective of the strategic choices of cluster firms, their decision-making exhibits a threshold effect. Theoretically, when $1+d_i < r$, individual rationality aligns with collective rationality. In this case, cluster firms adopting the investment strategy can not only maximise their individual benefits but also promote the overall benefits of the industrial cluster. When $1 < r < 1+d_i$, individual rationality and collective rationality diverge. Cluster firms tend to adopt the non-investment strategy

to maximise individual benefits; however, from the perspective of the industrial cluster as a whole, collective benefit is optimised only when all cluster firms adopt the investment strategy, creating a "prisoner's dilemma". When $r = 1 + d_i$, the system reaches a critical transition point in the game dynamics; this threshold marks the key juncture at which the strategic choices of cluster firms shift from being dominated by individual rationality to being dominated by collective rationality.

## Evolutionary rule

Evolutionary rules describe the behavioural logic whereby participants dynamically adjust their strategies in pursuit of higher payoffs, forming a core mechanism in complex network game analysis. Within the evolutionary game framework, the emergence and sustainability of cooperative behaviour are influenced not only by network structure but also, crucially, by the evolutionary rules themselves. Under the assumption of bounded rationality, individuals select and compare payoff differences with reference actors, iteratively updating their own strategies to progressively optimise their benefits. This strategy update mechanism is not only essential for capturing the fundamental dynamics of complex network evolution but also provides a vital perspective for understanding the emergence of cooperative behaviours such as co-construction and co-creation. Evolutionary game theory often draws on biological evolutionary strategies or social decision-making mechanisms to design evolutionary rules, such as replicator dynamics [78], Fermi learning [79], and the Moran process [80]. The digital inclusion ecosystem co-construction problem examined in this paper shares similarities with public goods games; however, learning mechanisms that are overly complex or tailored to specific environments often fail to establish meaningful real-world correspondences, potentially undermining the model's interpretability [81]. Therefore, considering both classical status and general applicability, we incorporate Fermi learning into the system of evolutionary rules.

Reflecting real-world conditions, this study enhances the Fermi learning algorithm based on the random surfer model. The core mechanism of Fermi learning involves a game participant randomly selecting a neighbour as a reference and comparing payoffs via the Fermi function to decide whether to imitate that neighbour's strategy [79]. The random surfer model, meanwhile, simulates the behavioural patterns of network users browsing web pages. This retrieval mechanism more closely aligns with the behavioural characteristics of firms selecting reference objects within an industrial cluster, where a firm may imitate the strategies of upstream or downstream neighbours or draw lessons from experiences across different domains. Therefore, integrating the exploration mechanism of the random surfer model into Fermi learning better captures the dynamic process of strategy updates among firms in industrial clusters, providing evolutionary rules that more realistically reflect the co-construction behaviours within the digital inclusion ecosystem. The specific rules are as follows:

After randomly setting the strategies for the initial round, only one firm updates its strategy per unit time.

A firm $i$ is randomly selected to update its strategy.

During the update, a reference firm is selected according to the random surfer model: with probability $\gamma$, a neighbour is randomly chosen as the reference, and with probability $1-\gamma$, a firm is randomly selected from the entire industrial cluster network $I$ as the reference.

The payoffs of the reference firm are compared, and the decision to imitate its strategy is made with probability $R_{i \leftarrow j}$, determined by the Fermi function, expressed as:

$$R_{i \leftarrow j} = \frac{1}{1 + exp(\frac{\pi_i - \pi_j}{k})}$$

(3)

Here, $R_{i \leftarrow j}$ denotes the probability that firm $p_i$ imitates firm $p_j$, $\pi_i$ and $\pi_j$ represent the cumulative payoffs of firms $p_i$ and $p_j$ respectively, and $k$ denotes the noise parameter. When $k \rightarrow 0$, firm $p_i$ tends towards rational decision-making, and the learning process becomes deterministic, entirely governed by payoffs. When $k \rightarrow \infty$, firm $p_i$ tends towards irrational decision-making, and the learning process becomes stochastic, entirely independent of the payoffs of both parties. Setting $k = 0.1$ indicates that the decision-making process of the participants is one of bounded rationality [82].

## Monte carlo method

The Monte Carlo method is a numerical computation approach based on probability and statistics, whose essence lies in constructing stochastic sampling processes to approximate the behavioural characteristics of complex systems. By generating a large number of random samples and leveraging the law of large numbers, this method achieves statistical estimation of the target problem. It is particularly suitable for situations where systems are too complex to obtain analytical solutions, or where the problem itself lacks a closed-form analytical expression [83]. In this study, due to the complex characteristics of the industrial cluster digital inclusion ecosystem—involving multi-agent interactions, nonlinear feedback, and high stochasticity—traditional analytical methods struggle to capture the system's dynamic evolution process.

To better simulate real-world scenarios, the modelling introduces multiple stochastic mechanisms: firstly, randomness in the generation of the small-world network topology; secondly, randomness in the selection of connection targets between the opinion leader node and nodes in the small-world network; thirdly, randomness in the initial strategy set; fourthly, randomness in the selection of agents for Fermi learning updates under asynchronous updating; and fifthly, randomness in the selection of reference objects for Fermi learning. These stochastic elements can influence game outcomes and may even lead to extreme scenarios. Therefore, this study employs the Monte Carlo method, conducting a large number of independent repeated experiments to eliminate the interference of randomness and ensure the robustness and reliability of the research conclusions.

Each iteration through Fermi learning calculates the proportion of cluster firms adopting the investment strategy within the network, denoted as $\theta(x_t)$. To ensure data comparability, the platform enterprise acting as the opinion leader is excluded from the calculation of $\theta(x_t)$. After Monte Carlo simulation, the average proportion of cluster firms adopting the investment strategy—i.e., the investors—relative to all cluster firms, denoted as $\theta_a$, is calculated as follows:

$$\theta_a = \frac{1}{T} \sum_{t=1}^{T} \theta(x_t)$$

(4)

Here $x$ represents the number of game iterations, $t$ denotes the $t$-th Monte Carlo simulation, and $T$ is the total number of Monte Carlo simulations. According to the law of large numbers and moment estimation theory, when the sample size $T$ tends towards infinity, the sample average in Monte Carlo simulations will converge to the expected value. The Monte Carlo Step is an important temporal unit for measuring the progress of the game evolution. Within one Monte Carlo Step, the system typically executes a number of game iterations equal to the number of nodes in the network. This ensures, in a statistical sense, that each cluster firm node has at least one opportunity to be selected as the agent for Fermi learning within that step. Finally, the game process is illustrated in Fig 1.

## Simulation and analysis

### Network generation and measurement of opinionleadership

To derive valid and stable conclusions, parameters were set based on established literature. Small-world networks were generated with configurations of $n$= 20, σ= 4 [37], and $n$= 300, σ= 16 [84], employing a rewiring probability of $q$= 0.05 [84]. The evolutionary process involved $T$= 100 [85] independent Monte Carlo simulations. Ultimately, we will construct two types of networks as shown in Table 1.

To enhance the comparability and controllability of the simulated data, this study adopted a progressive approach to construct the "small-world + opinion leader" complex network. Specifically, the initial network structure was formed by first generating a small-world network for the industrial cluster based on the Watts-Strogatz model, into which a single platform enterprise, designated as the opinion leader, was embedded. The number of connections between this platform enterprise and the cluster firms was then incrementally increased until it was ultimately connected to all firms. This progressive

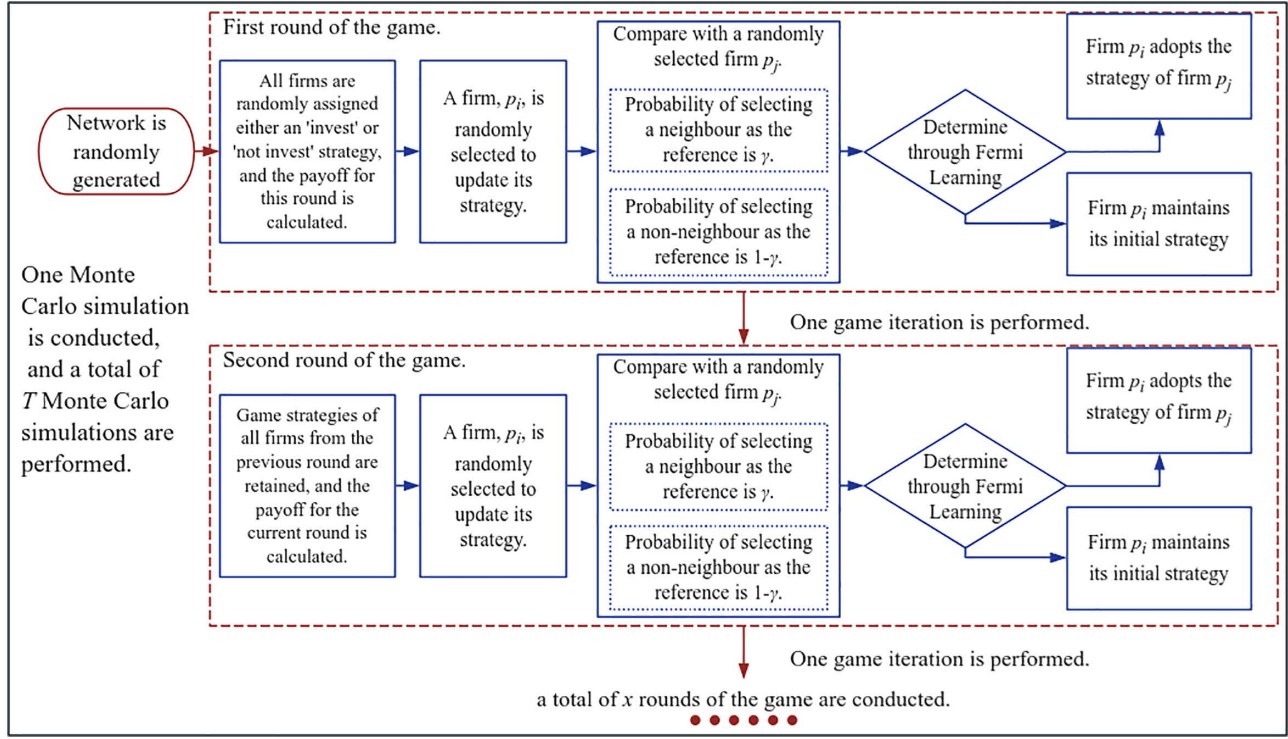

**Fig 1. Flowchart of the game procedure.**

**Table 1. Two types of small-world networks.**

| Type | n values | σ values | q values |
|---|---|---|---|
| Network Type 1 | 20 | 4 | 0.05 |
| Network Type 2 | 300 | 16 | 0.05 |

construction method offers a dual advantage. On one hand, it preserves the potential randomness inherent in real-world networks through the introduction of stochastic connection mechanisms. On the other hand, by systematically controlling the increase in connection numbers, it imbues the network embedding process of the opinion leader with observable regularity, thereby significantly enhancing data comparability across different simulation scenarios and providing a reliable network foundation for subsequent evolutionary game analysis. Fig 2 illustrates the establishment process for a "small-world + opinion leader" complex network with parameters $n= 20$, $\sigma= 4$. An equivalent procedure was used to generate a network with $n= 300$, $\sigma= 16$.

In Fig 2, the *CR* (Connection Ratio) metric quantifies the extent of connectivity between the platform enterprise and the cluster firms. It is defined as the proportion of cluster firms that have a direct connection to the platform enterprise relative to the total number of cluster firms, with $CR \in [0, 1]$. For instance, $CR= 0.5$ indicates that 50% of the cluster firms are directly connected to the platform enterprise. Node 21 within the network, i.e., the $p_{n+1}$, represents the platform enterprise, embedded into the small-world network as the opinion leader. To assess the network effects attributable to this opinion leader, a "No Platform" stage was established as a control scenario, representing the situation where the platform enterprise is not connected to the cluster network. The PR values for the four stages depicted in Fig 2 were sequentially

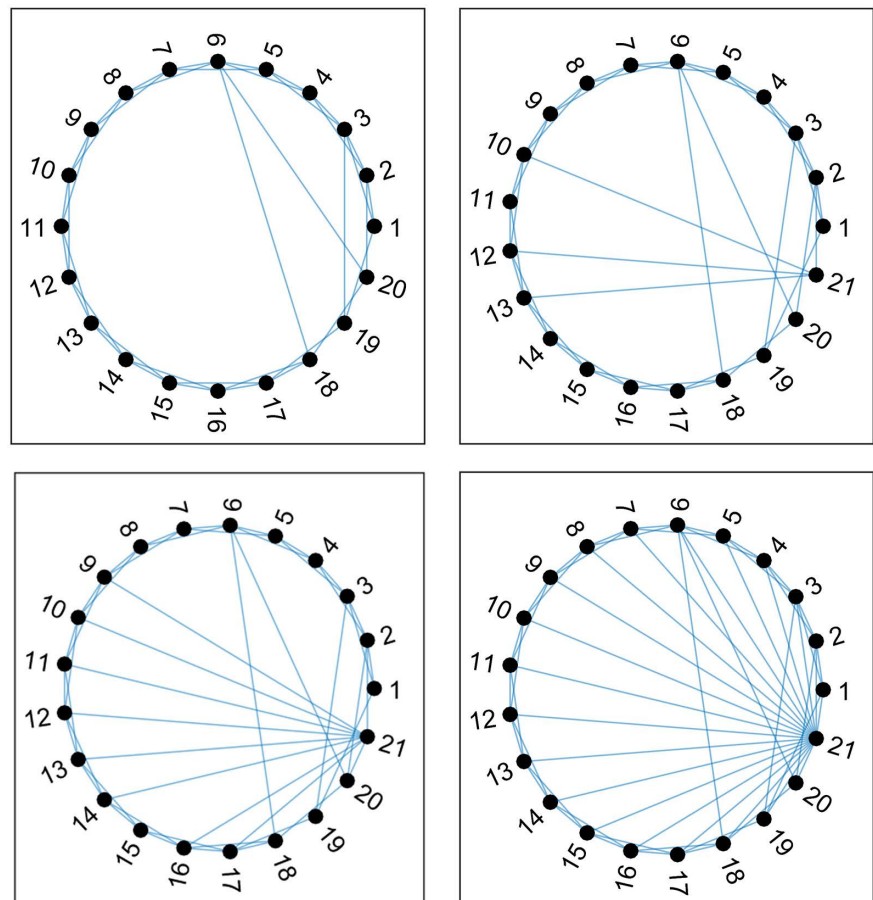

**Fig 2. The establishment process for a "small-world + opinion leader" complex network with n= 20 and $\sigma$ = 4. (a)** Stage 1: No Platform, **(b)** Stage 2: CR=$\sigma$/n, **(c)** Stage 3: CR= 0.5, **(d)** Stage 4: CR= 1.

calculated. After performing 100 Monte Carlo simulations, the average PR values for all nodes in the two types of "small-world + opinion leader" complex networks were obtained, as presented in Table 2.

As shown in Table 2, as CR progressively increases, the *PR* of the platform enterprise (node *n*+1) exhibits significant growth, indicating its rising importance within the network and the increasing prominence of its opinion leadership status. Specifically, in the baseline "No Platform" scenario, the *PR* value for the platform enterprise is zero, confirming the network's complete independence from this node. When *CR*=$\sigma$/n, the average *PR* value of the platform enterprise is similar to that of other nodes, suggesting its influence is not yet pronounced. At *CR*= 0.5, the average *PR* of the platform enterprise reaches the maximum among all nodes in both complex network types, signifying that this node possesses the characteristics of an opinion leader. Finally, when *CR*= 1, indicating the $p_{n+1}$ node is connected to all other nodes, the PR value of the platform enterprise approaches its theoretical maximum, fully demonstrating its central, hub-like role in co-building the digital inclusion ecosystem.

## Impact of public goods game variables on evolution

Within the public goods game model, the number of players, the return on investment coefficient, and the investment cost constitute key variables determining the evolutionary outcome. Due to the heterogeneity in the number of neighbours

**Table 2. Average *PR* for nodes in the "small-world + opinion leader" complex network.**

| Network Type | Stage | Max Avg PR | Avg PR of All Nodes | Avg PR of Each Node | | | | |
|---|---|---|---|---|---|---|---|---|
| | | | | $p_{n+1}$ | ... | $p_4$ | $p_3$ | $p_2$ | $p_1$ |
| Network Type 1: $n$= 20, $\sigma$= 4 | No Platform | 0.050 69 | 0.05 | 0 | ... | 0.049 76 | 0.050 26 | 0.050 14 | 0.049 26 |
| | $CR=\sigma/n$ | 0.048 80 | 0.047 61 | 0.044 55 | ... | 0.047 08 | 0.047 99 | 0.047 59 | 0.047 33 |
| | $CR$= 0.5 | 0.090 69 | 0.047 61 | 0.090 69 | ... | 0.045 08 | 0.045 51 | 0.045 26 | 0.045 43 |
| | $CR$= 1 | 0.151 53 | 0.047 61 | 0.151 53 | ... | 0.042 26 | 0.042 59 | 0.042 51 | 0.041 91 |
| Network Type 2: $n$= 300, $\sigma$= 16 | No Platform | 0.003 37 | 0.003 33 | 0 | ... | 0.003 33 | 0.003 32 | 0.003 31 | 0.003 32 |
| | $CR=\sigma/n$ | 0.003 47 | 0.003 32 | 0.003 28 | ... | 0.003 36 | 0.003 32 | 0.003 32 | 0.003 31 |
| | $CR$= 0.5 | 0.025 39 | 0.003 32 | 0.025 39 | ... | 0.003 25 | 0.003 23 | 0.003 23 | 0.003 24 |
| | $CR$= 1 | 0.048 11 | 0.003 32 | 0.048 11 | ... | 0.003 17 | 0.003 16 | 0.003 15 | 0.003 16 |

among firms within the industrial cluster, the number of game participants for firm $i$ can be expressed as $1+d_i$. To comprehensively investigate the return on investment effect, the following three scenarios are established: $r=\sigma/2$, $r=\sigma+1$ and $r=2\sigma$. Theoretically, When $r=\sigma/2$, the condition $1 <r< 1+d_i$ is satisfied, indicating that the investment return is insufficient to incentivise widespread cooperation. When $r=\sigma+1$, most cluster firms reach the critical value $r= 1+d_i$, placing the game dynamics at a phase transition point between cooperation and free-riding. When $r= 2\sigma$, cluster firms satisfy, creating favourable conditions for the evolution of cooperation. Concurrently, to examine the impact of investment costs, scenarios with $c= 10$ and $c= 100$ are set for comparative analysis. To ensure data comparability, this study uses the small-world network from the "No Platform" stage as the benchmark scenario, thereby excluding interference from the platform enterprise's involvement. After 100 Monte Carlo simulations, the game evolution results within 10 Monte Carlo steps are obtained, as shown in Fig 3.

The results indicate that the investment cost has no significant impact on the evolution, whereas increasing the return on investment coefficient exhibits an incentivising effect. Although the initial strategies of the firms are random, after 100 Monte Carlo simulations, the investor ratio $F_a$ converges to 0.5, meaning the number of initial investors equals the number of non-investors. As the value of $r$ increases, cluster firms become incentivised by the higher investment returns, showing

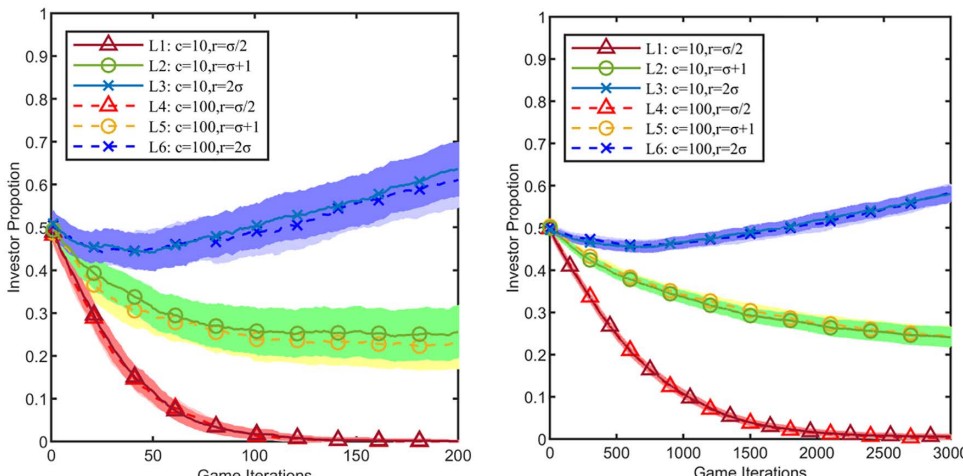

**Fig 3. Impact of public goods game variables on evolution.** The light-colored shaded areas represent the 95% confidence intervals corresponding to the line charts. **(a)** Network Type 1, **(b)** Network Type 2.

a significantly stronger tendency to choose the investment strategy. As illustrated in Fig 3, both the smaller-scale (Network Type 1) and larger-scale small-world networks (Network Type 2) exhibit similar evolutionary patterns. After the investment cost changes from $c$= 10 to $c$= 100, the evolutionary curves for the same rate of return overlap, indicating no significant impact on the evolutionary outcome under either network scale. Notably, when $r$= $2\sigma$ exceeds the critical value $\sigma$+1, the proportion of investors rises and surpasses that of non-investors. This implies that the overall willingness to "invest" within the industrial cluster is greater than the willingness "not to invest". Conversely, when the $r$ value reaches the critical value $\sigma$+1, the proportion of investors decreases and falls below that of non-investors. This suggests that the overall willingness "not to invest" within the industrial cluster is greater than the willingness to "invest". This observation deviates from theoretical expectations, potentially due to the influence of network structure on the results. When $r$=$\sigma$/2 falls below the critical value $\sigma$+1, free-riding behaviour becomes prevalent among cluster firms.

### Impact of platform enterprises' connection ratio on evolution

According to the analytical results in Table 2, within the same network type, as the platform enterprise increases its business interactions with the industrial cluster network, its PR value rises significantly, indicating its growing importance within the network. Since the investment cost shows no obvious impact on evolution, the following analysis focuses on the scenario where c = 10 for simplification. For the two network types, n = 20, $\sigma$ = 4 and n = 300, $\sigma$ = 16, and under the three return coefficients r = $\sigma$/2 + 1, r = $\sigma$ + 1, r = $2\sigma$+1, the number of investors in the complex network is positively correlated with the total cluster payoff. When all firms choose the investment strategy, the industrial cluster's payoff is optimised. Since maximising the industrial cluster's benefit is the goal of the platform enterprise as a "public agent", it consistently maintains the investment strategy. To ensure data comparability, the investment behaviour of the platform enterprise is excluded from the $F_a$. After 100 Monte Carlo simulations, the impact of the platform enterprise's connection ratio on the evolutionary process within 10 Monte Carlo steps is shown in Fig 4.

The results demonstrate that the "exemplary role" of the platform enterprise strengthens as it establishes connections with more cluster firms. As shown in Fig 4, the two network types exhibit similar evolutionary outcomes. As the connection ratio $CR$ progressively increases from 0 to 1, the strategic evolution trajectory of the industrial cluster gradually deviates from the benchmark "No Platform", reaching its maximum influence intensity when $CR$= 1. Specifically, under Conditions 1 and 2 ($r$=$\sigma$/2 + 1), where collective rationality conflicts with individual rationality, cluster firms tend to adopt the "not invest" strategy. Although this tendency persists as the connection ratio $CR$ rises, its strength gradually weakens. Under Conditions 3 and 4 ($r$=$\sigma$+1), where the game dynamics reach the theoretical critical point, cluster firms exhibit a slight tendency towards free-riding. However, as the connection ratio $CR$ increases, this tendency gradually diminishes. Under Conditions 5 and 6 ($r$= $2\sigma$+1), where individual rationality aligns with collective rationality, cluster firms tend to adopt the "invest" strategy. As the connection ratio $CR$ rises, this "invest" tendency is reinforced among cluster firms.

## Further research

### Impact of platform enterprise strategies on evolution

Based on the results in Fig 3, when the platform enterprise establishes direct connections with all firms in the industrial cluster ($CR$= 1), it fully attains the status of an opinion leader, and its influence on the strategic evolution of the cluster reaches its maximum. Therefore, to highlight the impact of the platform enterprise's strategy on the evolutionary process, the $CR$ value is set to 1. To ensure data comparability, the investment behaviour of the platform enterprise is excluded from the $F_a$. With fixed parameters $c$= 10 and $T$= 100, the impact of the platform enterprise's strategy on the evolutionary process is examined under three return coefficient scenarios: $r$=$\sigma$/2 + 1, $r$=$\sigma$+1 and $r$= $2\sigma$+1, as shown in Fig 5.

In Fig 5, "Platform Investment" indicates that the platform enterprise consistently adopts the investment strategy; "Platform Free-rides" indicates that it consistently adopts the non-investment strategy; "Platform Adjusts Strategy" indicates that the platform enterprise dynamically adjusts its strategy based on the Fermi learning algorithm improved with the

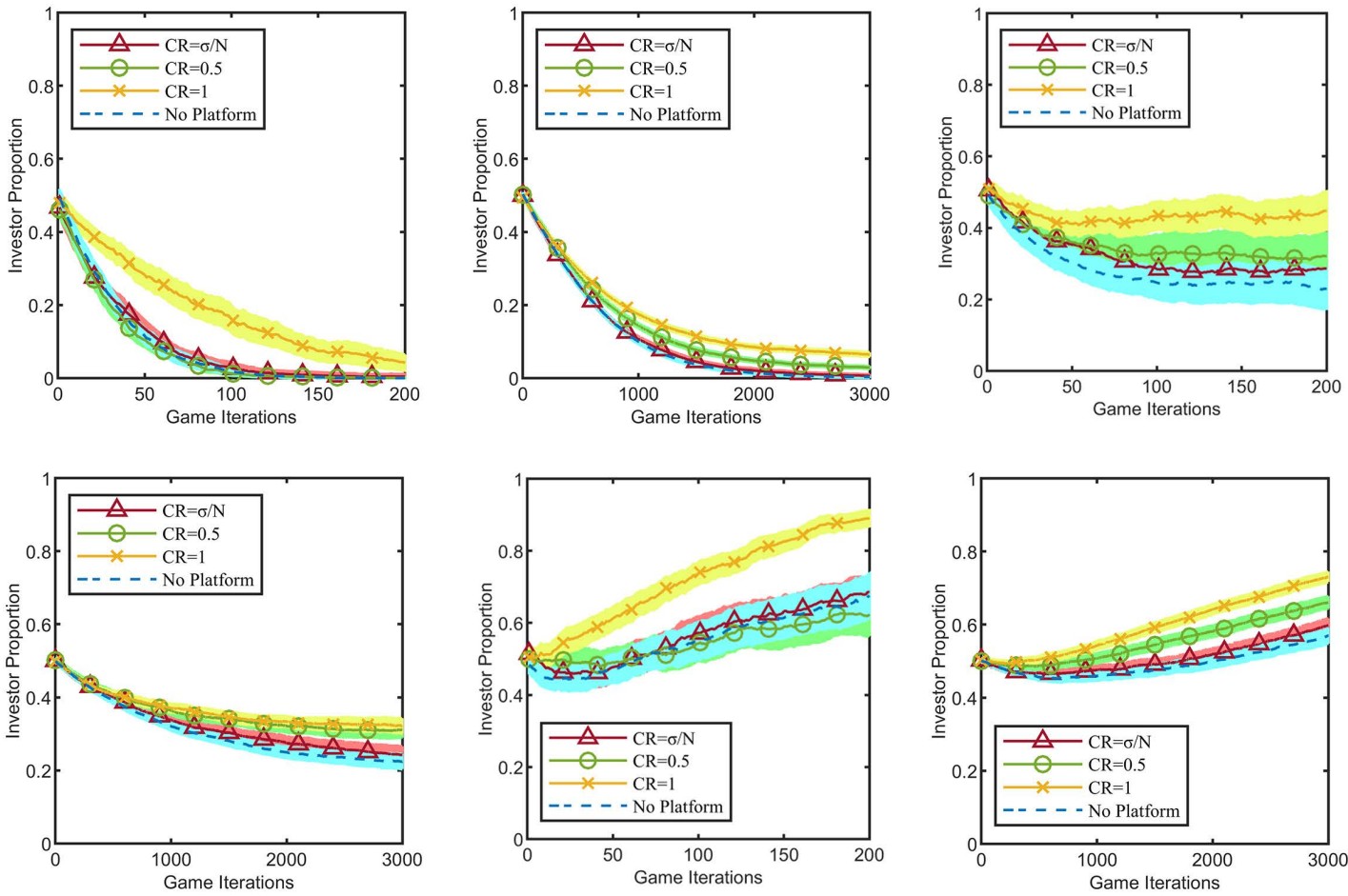

**Fig 4. Impact of the opinion leader's connection ratio on evolution.** The light-colored shaded areas represent the 95% confidence intervals corresponding to the line charts. **(a)** Condition 1: Network Type 1 and **r**=σ/2 + 1, **(b)** Condition 2: Network Type 2 and **r**=σ/2 + 1, **(c)** Condition 3: Network Type 1 and **r**=σ+1, **(d)** Condition 4: Network Type 2 and **r**=σ+1, **(e)** Condition 5: Network Type 1 and **r**= 2σ+1, **(f)**Condition 6: Network Type 2 and **r**= 2σ+1.

random surfer model; and "No Platform" indicates that the platform enterprise is not embedded in the cluster and has no business connections with the cluster firms.

The study demonstrates that the platform enterprise maintaining an investment strategy yields the best incentivising effect on the co-construction of the digital inclusion ecosystem by the industrial cluster. As shown in Fig 5, across all six conditions, the investor ratio under the "Platform Investment" strategy is higher than that in the "No Platform" control group. Conversely, under all six conditions, the investor ratio under the "Platform Free-rides" strategy is lower than that in the "No Platform" control group, indicating that free-riding by the platform enterprise is detrimental to the co-construction of the digital inclusion ecosystem. Under Conditions 1 and 2, "Platform Adjusts Strategy" has an incentivising effect on co-construction, with the investor proportion under "Platform Adjusts Strategy" being higher than the "No Platform" control group. Under Conditions 3–6, however, "Platform Adjusts Strategy" is detrimental to co-construction, as the investor proportion under "Platform Adjusts Strategy" gradually falls below that of the "No Platform" control group. Notably, across all six conditions, the investor proportion under the "Platform Investment" strategy exceeds that observed under other strategies. However, under Conditions 1 and 2, none of the platform enterprise's strategies, including maintaining investment, are able to reverse the declining trend in the investor proportion.

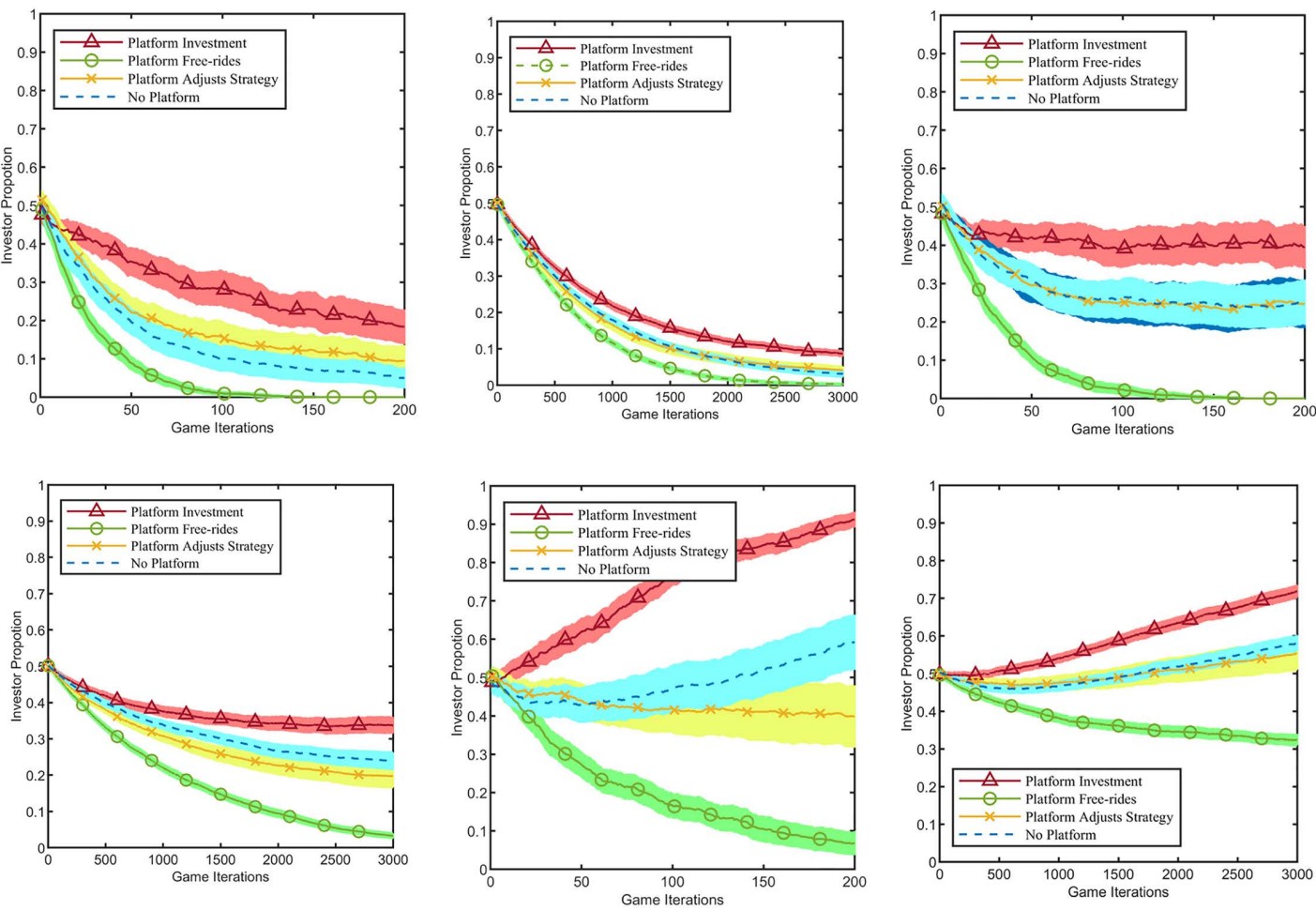

**Fig 5. Impact of platform enterprise strategies on evolutionary dynamics.** The light-colored shaded areas represent the 95% confidence intervals corresponding to the line charts. **(a)** Condition 1: Network Type 1 and **r**=$\sigma$/2 + 1, **(b)** Condition 2: Network Type 2 and **r**=$\sigma$/2 + 1, **(c)** Condition 3: Network Type 1 and **r**=$\sigma$+1, **(d)** Condition 4: Network Type 2 and **r**=$\sigma$+1, **(e)** Condition 5: Network Type 1 and **r**= 2$\sigma$+1, **(f)** Condition 6: Network Type 2 and **r**= 2$\sigma$.

## Impact of rewards, punishments, and collective agreements on evolution

Within the digital inclusion ecosystem, the dominant power of leading enterprises enables them to shape and direct collective behaviour patterns. The power structure within an ecosystem not only influences members' behavioural choices but can also generate positive synergistic effects [86]. Leading enterprises primarily provide platform infrastructure and establish fair, transparent participation rules, through which they acquire, maintain, and exercise their advantageous power position [87]. This structural asymmetry in power favours the orchestration of resources within their internal ecosystem by the leading enterprise [88]. For instance, exercising platform-based rewards/punishments and establishing collective agreements can enhance relational reciprocity among members and foster the construction of trust mechanisms, thereby reducing the incidence of opportunistic behaviour [89].

The results from Conditions 1 and 2 in Fig 5 indicate that, under the given parameter settings, none of the platform enterprise's strategies can reverse the declining trend in the investor proportion. To highlight the impact of governance mechanisms—specifically, the platform's exercise of reward power, punishment power, and the formation of collective agreements—on the evolutionary process, this study selects Conditions 1 and 2 as the baseline environments. We set

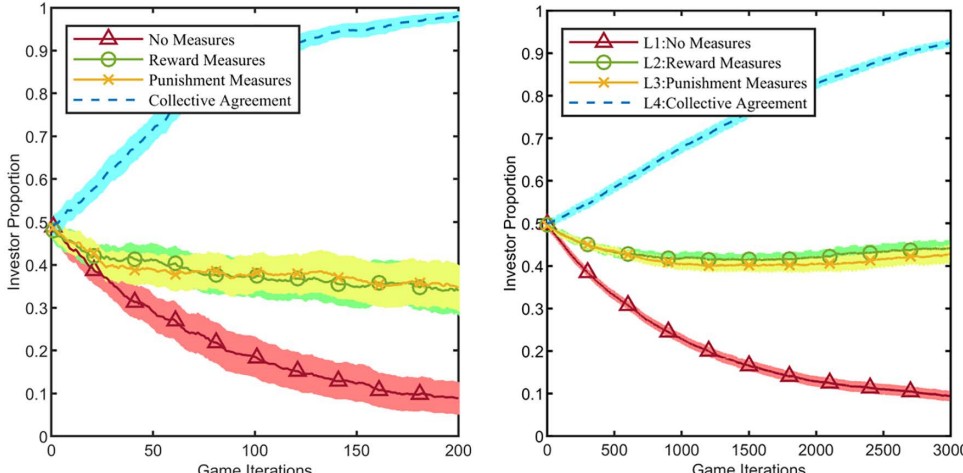

**Fig 6. Impact of rewards, punishments, and collective agreements on evolution.** The light-colored shaded areas represent the 95% confidence intervals corresponding to the line charts. **(a)** Condition 1: Network Type 1 and **r**=σ/2 + 1, **(b)** Condition 2: Network Type 2 and **r**=σ/2 + 1.

CR = 1 and $\lambda_{n+1}$= 1, meaning the platform enterprise exerts maximum influence and consistently maintains its investment strategy. It is crucial to note that the effective exercise of the platform's governance power necessitates its embeddedness within the industrial cluster. Therefore, the "No Platform" scenario, where the platform is detached from the cluster, is not considered. To ensure comparability of findings, parameters remain fixed at c = 10, T = 100. The platform enterprise consistently maintains its investment strategy throughout the game, and the evolutionary process over 10 Monte Carlo steps is shown in Fig 6.

Fig 6 compares the effects of different governance measures implemented by the platform enterprise on the co-construction of the digital inclusion ecosystem. Here, "No Measures" represents the baseline scenario where the platform implements no governance powers; "Reward Measures" refers to the platform providing a reward equivalent to 50% of the investment cost to those cluster firms (with which it has business connections) that adopt the investment strategy; "Punishment Measures" denotes the platform imposing a fine equivalent to 50% of the investment cost on associated firms that choose to free-ride; "Collective Agreement" signifies a binding cooperative agreement reached between the platform and all cluster firms. This agreement requires all signatory firms, including the platform, to pre-pay the investment cost, strictly earmarking these funds solely for digital inclusion ecosystem construction. If a firm does not utilise its pre-paid investment cost within the cycle, the unused funds are redistributed equally among all signatory firms to the agreement.

The results indicate that the effectiveness of platform-based rewards and punishments in promoting co-construction is comparable, but the collective agreement yields the optimal outcome. As shown in Fig 5, in both the smaller-scale (Network Type 1) and larger-scale (Network Type 2) industrial cluster networks, rewards or punishments set at 50% of the investment cost can slow the trend of collective free-riding among cluster firms but cannot reverse this outcome. In our simulation environment, the collective agreement mechanism demonstrated the ability to reverse this trend, suggesting its potential as an effective institutional design.

## Conclusions, recommendations and limitations

### Conclusions

Based on the small-world network characteristics of industrial clusters, the opinion leadership status of platform enterprises, and the public-good attributes and externalities of the digital inclusion ecosystem, this research constructs a complex

network public goods evolutionary game model integrating "small-world + opinion leader" and incorporates the Fermi learning algorithm improved with the surfer model as the evolutionary rule. Through Monte Carlo simulations, we analyse the impacts of investment costs, return on investment coefficients, platform connection ratios, platform strategies, platform reward/punishment powers, and collective agreements on network evolution. The main conclusions are as follows:

Under the modeled conditions, the core driving mechanism for co-building the digital inclusion ecosystem in industrial clusters lies in the alignment of individual returns and collective rationality, with evolutionary outcomes heavily influenced by network structure and the rate of return on investment. Simulation results reveal a significant "threshold effect" in the cooperative behaviour of cluster firms: when the return on investment coefficient $r$ surpasses the critical value defined by the number of game participants, individual rationality aligns with collective rationality, enabling cooperation to emerge. Conversely, when $r$ falls below this threshold, the system descends into a typical "Olson's dilemma," where free-riding becomes the dominant individual strategy. This finding clarifies that the fundamental driver for co-building the digital inclusion ecosystem is not corporate altruism, but rather meticulously designed incentive mechanisms that ensure cooperative behaviour yields tangible, anticipated excess returns for firms.

The "exemplary role" of the platform enterprise as an opinion leader is significant. Its influence strengthens with increased network connectivity, yet its strategic choices exert a "dual-directional regulatory effect" on system evolution. This study finds that the platform's connection ratio is crucial to its influence. When $CR= 1$, meaning the platform connects to all cluster firms, its opinion leadership is most, and its ability to guide group strategy peaks. More importantly, the platform's own strategy serves as a powerful signal: the "Platform Investment" strategy effectively incentivises cooperation and increases the proportion of investors cluster-wide. In contrast, the "Platform Free-rides" strategy creates a strong negative demonstration effect, potentially pushing otherwise cooperative scenarios towards collapse. This indicates that the platform enterprise is not merely a technological intermediary but also a behavioural benchmark; the trade-off between its public-agent and economic-agent attributes directly shapes the trajectory of ecological co-construction.

To maximise the platform's positive role and curb its negative impact in practice, the key lies in constraining its opportunistic behaviour and guiding it to exercise "platform power" for effective governance.The research shows that if the platform adopts a "variable strategy" focused solely on short-term self-interest, the resulting uncertainty undermines the cluster's stable expectations, performing worse than the "No Platform" baseline. Therefore, external regulation and internal governance are necessary to compel the platform enterprise to honour its "maintain investment" commitment. Building on this, the platform can leverage its power as the ecosystem leader, utilising governance tools like rewards (for cooperators) and punishments (for free-riders) to significantly slow the decay of cooperation. This provides clear guidance for regulators and business practices: platforms must be both "restrained" from harmful actions and "empowered" to facilitate positive outcomes.

The most effective top-level design for avoiding cooperation dilemmas is establishing a "collective agreement" mechanism that incorporates binding commitments and benefit redistribution.A particularly encouraging finding is that while traditional reward and punishment measures merely slow but cannot reverse the free-riding trend, the "collective agreement" demonstrates a powerful capacity to promote cooperation. This mechanism, which requires signatory firms to pre-pay investment costs and stipulates the redistribution of unused funds, creates effects akin to "commitment devices" and an "interest community." It fundamentally alters firms' profit expectations, successfully rescuing the system from the "prisoner's dilemma." This insight advises policymakers and industrial cluster planners that fostering mandatory and reciprocal internal cluster agreements is a fundamental institutional safeguard for resolving the conflict between individual and collective rationality and achieving the sustainable development of the digital inclusion ecosystem.

### Recommendations

Based on the above findings, and to promote the collaborative construction of the digital inclusion ecosystem within industrial clusters while overcoming the free-rider cooperation dilemma, this paper proposes the following managerial insights and policy recommendations for governments, platform enterprises, and the broader cluster firms, respectively:

First, for governments and regulatory bodies, whose core role is that of "rule-setter and environment enabler."Governments should focus on top-level institutional design rather than direct market intervention. Primarily, they should actively advocate for and lead the organisation of "collective agreements" related to digital inclusion within industrial clusters, providing a legal framework and performance guarantees to make this mechanism operational and binding. Secondly, they should implement "targeted incentives" policies, using tools such as tax relief and specific subsidies to compensate firms—especially SMEs near the cooperation threshold—that bear the costs of ecological co-construction but see their positive externalities captured by others. This effectively enhances their return on investment, enabling them to cross the cooperation threshold. Finally, they should strengthen the guidance and supervision of platform enterprise behaviour by establishing a "digital inclusion" performance evaluation system. This would reward platforms that fulfil their social responsibilities and penalise speculative behaviours such as free-riding and abusing market dominance, thereby compelling them to wield their power for the public good.

Second, for platform enterprises, whose core role is that of "ecosystem coordinator and value co-creation leader."Platform enterprises must recognise their dual identity as both "economic agents" and "public agents," understanding that their long-term interests are deeply intertwined with the healthy development of the industrial cluster. Firstly, they must steadfastly adhere to the long-term strategic commitment of "maintaining investment," using their undisputed opinion leader status to set a cooperative benchmark for the cluster through a "demonstration-imitation" mechanism. Secondly, they should proactively transform their network connectivity advantages into governance strengths, actively exercising "platform power" to design and implement transparent and fair reward and punishment mechanisms. This includes preferential access to traffic, data, or technical support for cooperators and necessary restrictions on ecosystem access for blatant free-riders. Lastly, they should voluntarily open core interfaces of their digital infrastructure to lower the technical and cost barriers for SMEs integrating into the ecosystem, thereby stimulating participation motivation from the supply side.

Third, for the wider cluster firms (especially SMEs), whose core role is that of "active participant and capability builder."-Cluster firms should abandon short-term speculative mindsets and strategically recognise that integrating into the digital inclusion ecosystem is a crucial path to overcoming their own resource constraints and achieving sustainable development. Firstly, they should actively respond to collective agreements and governance rules initiated by governments and platform enterprises, positioning themselves as "interest communities" within the ecosystem rather than "bystanders," seeking shared benefits through co-construction. Secondly, they should proactively leverage the empowerment opportunities provided by platforms, increasing investment in digital skills training and new technology application to address digitalisation shortcomings and enhance their capacity to absorb and utilise the ecosystem's spillover value. Finally, firms should strengthen peer-to-peer learning and exchange within the industrial cluster network, fostering a community culture that glorifies cooperation and shuns isolation, thereby using informal norms to supplement formal institutions and collectively curb opportunistic behaviour.

## Limitations

Whilst this study provides valuable insights into the orchestrating role of platform enterprises in digital inclusion, several limitations should be acknowledged to guide future research.

First, the model simplifies the complex reality of industrial clusters by representing them with small-world network topologies. Although this structure captures high clustering and short path lengths, real-world clusters may exhibit more heterogeneous and dynamic network formations, including scale-free properties or multi-layer networks, which were not explored here. Future work could incorporate more diverse and evolving network structures to enhance the model's realism.

Second, the model assumes homogeneity among cluster firms in terms of their cost structures and learning capabilities. In practice, firms vary significantly in size, resources, and absorptive capacity, factors that likely influence their strategic choices in digital inclusion efforts. Introducing firm heterogeneity into the model parameters would provide a more nuanced understanding of the ecosystem's evolution.

 

Third, the study focuses primarily on the platform's role as an opinion leader and the effects of its connection ratio, without deeply examining variations in its intrinsic attributes, such as its reputation, technological sophistication, or the specific design of its governance mechanisms. Different types of platform enterprises might exert varying levels of influence, a dimension that warrants further investigation.

Finally, although the Monte Carlo simulation helps mitigate randomness, the model's parameters—such as the specific values of the return coefficient were set based on existing literature to ensure comparability rather than being empirically estimated from real-world data. Future research could benefit from calibrating these parameters with empirical data from specific industrial clusters to improve the model's external validity and predictive power.

Despite these limitations, the findings offer a robust theoretical foundation and highlight the critical role of institutional arrangements like collective agreements. Addressing these constraints in future studies will further refine our understanding of digital inclusion dynamics in industrial clusters.

## Supporting information

**S1 Appendix. Code and data.** The code and data have both been uploaded to OSF. https://osf.io/axtf9/overview?view_only=f39e90a6a71e40cd811aa5469a4aa9f6.
(ZIP)

**S2 Table. List of parameters.**
(PDF)

## Acknowledgments

Thanks to Guangxi City Vocational University for providing computing power support for this research.

## Author contributions

**Conceptualization:** Dingteng Wang.

**Data curation:** Dingteng Wang.

**Formal analysis:** Dingteng Wang.

**Investigation:** Dingteng Wang, Ningning Li, Guoan Liang.

**Project administration:** Shengming Li.

**Resources:** Ningning Li, Guoan Liang.

**Supervision:** Shengming Li.

**Validation:** Dingteng Wang.

**Writing – original draft:** Dingteng Wang.

**Writing – review & editing:** Ningning Li, Guoan Liang.

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
