## [Decision Letter · Decision Letter 0]

30 Dec 2025

PONE-D-25-58355The Orchestrating Role of Platform Enterprises in Digital Inclusion: A Network Game Perspective from Industrial Clusters

PLOS One

Dear Dr. Wang,

Thank you for submitting your manuscript to PLOS ONE. After careful consideration, we feel that it has merit but does not fully meet PLOS ONE’s publication criteria as it currently stands. Therefore, we invite you to submit a revised version of the manuscript that addresses the points raised during the review process.

Overall, both reviewers agree that the manuscript presents a methodologically sound and technically rigorous study.  The reviewers did not identify substantive concerns regarding data integrity, analytical validity, or compliance with the journal’s data availability requirements. The topic is timely and relevant, particularly given the growing importance of platform enterprises and digital inclusion within industrial cluster contexts.

That said, the reviewers have raised several important issues that must be addressed before the manuscript can be considered for publication. These concerns do not call into question the validity of the empirical or simulation results, but they do affect the conceptual clarity, interpretive coherence, and transparency of the study. .

The revisions are primarily aimed at strengthening conceptual clarity, interpretive precision, and methodological transparency.

We look forward to receiving your revised manuscript.

Kind regards,

Juan E. Trinidad-Segovia, PhD

Section Editor

PLOS One

Journal Requirements:

This research was supported by the Guangxi Science and Technology Think Tank Key Project (Grant No. K-88) from the Guangxi Association for Science and Technology, the Guangxi Basic Ability Improvement Project for Young and Middle-aged Teachers in Universities (Grant No. 2024KY1937) from the Department of Education of Guangxi Zhuang Autonomous Region, and the Guangxi Vocational Education Teaching Reform Research Project (Grant No. GXGZJG2024A004) from the Department of Education of Guangxi Zhuang Autonomous Region.

This work was supported by the Key Project of Guangxi Science and Technology Think Tank (Grant No. K-88), the Basic Ability Improvement Project for Young and Middle-aged Teachers in Guangxi Universities (Grant No. 2024KY1937), and the Guangxi Vocational Education Teaching Reform Research Project (Grant No. GXGZJG2024A004). And thanks to Guangxi City Vocational University for providing computing power support for this research.

This research was supported by the Guangxi Science and Technology Think Tank Key Project (Grant No. K-88) from the Guangxi Association for Science and Technology, the Guangxi Basic Ability Improvement Project for Young and Middle-aged Teachers in Universities (Grant No. 2024KY1937) from the Department of Education of Guangxi Zhuang Autonomous Region, and the Guangxi Vocational Education Teaching Reform Research Project (Grant No. GXGZJG2024A004) from the Department of Education of Guangxi Zhuang Autonomous Region.

Reviewers' comments:

Reviewer's Responses to Questions

**Comments to the Author**

1. Is the manuscript technically sound, and do the data support the conclusions?

Reviewer #1: Partly

Reviewer #2: Yes

2. Has the statistical analysis been performed appropriately and rigorously? 

Reviewer #1: Yes

Reviewer #2: Yes

3. Have the authors made all data underlying the findings in their manuscript fully available?

Reviewer #1: No

Reviewer #2: Yes

4. Is the manuscript presented in an intelligible fashion and written in standard English?

Reviewer #1: Yes

Reviewer #2: Yes

5. Review Comments to the Author

Reviewer #1: Dear authors,

This manuscript is characterized by a high level of theoretical depth and methodological soundness in its analysis of digital inclusion processes in industrial clusters. A significant strength of the work is its use of formalized tools from evolutionary game theory and network modeling, which allow us to identify dynamic mechanisms of coordination and simulation learning among economic agents. Particular attention is paid to the emphasis on the role of platform enterprises as structurally significant nodes of network interaction, as well as to the analysis of institutional mechanisms for overcoming collective action problems. The relevance of this study stems from the growing importance of the platform economy and the need to develop theoretically sound approaches to managing the digital transformation of cluster systems, making the findings relevant for further academic discussions.

At the same time, to enhance the scientific transparency and reproducibility of the results, the work requires a number of clarifications. In particular, it seems advisable to more clearly articulate the model nature of the data used and limit the interpretation of the obtained results to the framework of theoretical and simulation analysis. The methodological section could be strengthened by a more detailed and structured description of the modeling algorithm, including formalization of the simulation stages, parameters, and scenarios used. Clarification of these aspects would simplify subsequent verification of the model and improve its applicability for further research.

Reviewer #2: Dear Authors,

Thank you for the opportunity to review your manuscript. I appreciate the effort invested in this study and the care taken in the empirical design. Overall, the manuscript presents a technically sound piece of research, and the data and analytical methods appear appropriate and competently applied to the research questions posed.

From a technical standpoint, the study is well executed. The methodological approach is clearly described, the statistical analyses are performed rigorously, and the conclusions are generally supported by the results presented. I did not identify issues related to data quality, analytical validity, or compliance with data availability requirements.

The main area where the manuscript would benefit from revision concerns conceptual clarity and interpretive coherence, rather than methodological soundness. The paper departs from an interesting and potentially valuable theoretical starting point; however, this point is not developed consistently throughout the manuscript. Several conceptual perspectives and ideas are introduced across the introduction and discussion, but their respective roles are not always clearly delimited. As a result, the argument occasionally shifts between frameworks without making explicit how they relate to one another or which perspective is guiding the interpretation of the findings.

Importantly, this issue does not undermine the validity of the empirical analysis itself. Rather, it affects how the results are framed and how readers, particularly those from different disciplinary backgrounds, will understand the contribution of the study. Clarifying the primary conceptual lens (or lenses), defining key concepts more explicitly, and ensuring consistent use of those concepts across sections would substantially improve the readability and interpretability of the manuscript.

With respect to presentation and language, the manuscript is generally intelligible and written in acceptable standard English. Some sections, especially in the introduction and discussion, would benefit from minor language polishing and clearer signposting to guide the reader through the conceptual argument. These are issues of clarity and organization rather than grammatical correctness.

My comments are intended to be focused and realistic. Addressing them does not require changes to the data or analytical strategy, but rather a tightening of the conceptual framing and discussion to better align ideas, methods, and conclusions.

In summary, I consider the manuscript to be methodologically sound and suitable for publication following revision, and I hope these comments are helpful in strengthening the clarity and impact of the work.

Kind regards,

Reviewer

6. PLOS authors have the option to publish the peer review history of their article (what does this mean? ). If published, this will include your full peer review and any attached files.). If published, this will include your full peer review and any attached files.

**Do you want your identity to be public for this peer review?** For information about this choice, including consent withdrawal, please see our For information about this choice, including consent withdrawal, please see our Privacy Policy ..

Reviewer #1: No

Reviewer #2: **Yes:** Dr. Rudy Fernandez-EscobedoDr. Rudy Fernandez-Escobedo

---

## [Author Response · Author response to Decision Letter 1]

7 Jan 2026

Dear Editors and Reviewers,

Thank you for handling our manuscript and for providing highly constructive feedback. We have carefully reviewed and considered all suggestions and have undertaken a systematic revision of the manuscript accordingly. Below, we provide a point-by-point response detailing the changes made.

1. Responses to Journal Formatting and Policy Requirements

1.1 Formatting and File Naming

We have carefully formatted the manuscript according to the PLOS ONE template and have prepared the three required files:

"Response to Reviewers"

"Revised Manuscript with Track Changes"

"Manuscript" (Clean version)

1.2 Code Sharing

The simulation code for this study was developed in MATLAB (version 2022b or later). To ensure reproducibility, the code has been uploaded to the Open Science Framework (OSF) and is available at the following link:

https://osf.io/axtf9/overview?view_only=f39e90a6a71e40cd811aa5469a4aa9f6

Detailed run instructions are provided. The code has also been uploaded as supplementary material to the PLOS ONE submission system.

1.3 Data Availability Statement

The Data Availability Statement has been updated to confirm that all simulation data and code are accessible via the OSF repository. The data underlying the figures have been provided as Supporting Information files. Furthermore, the simulation data files have been uploaded to the PLOS ONE submission system. The .fig files can be opened in MATLAB to view the detailed data.

1.4 Financial Disclosure Statement

The Guangxi Science and Technology Think Tank Key Project (Grant No. K‑88) funded computational resource leasing.The Guangxi Basic Ability Improvement Project for Young and Middle‑aged Teachers in Universities (Grant No. 2024KY1937) covered publication-related expenses.The Guangxi Vocational Education Teaching Reform Research Project (Grant No. GXGZJG2024A004) supported costs associated with literature retrieval, access, and research-related travel.The funders had no role in study design, data collection and analysis, decision to publish, or preparation of the manuscript.

1.5 Acknowledgments Section

All funding-related acknowledgments have been removed from this section, retaining only thanks to the institution that provided computational support.

1.6 No specific citations are required.

The reviewers' comments do not include recommendations to cite specific previously published works.

2. Response to Reviewer #1

We thank the reviewer for their valuable comments aimed at enhancing scientific transparency, reproducibility, and the clarity and structure of the methodological description.

In direct response:

All model code and simulation data have been made publicly available on OSF (link provided in Section 1.2).The same code and data have also been uploaded as supplementary files to the PLOS ONE submission system for convenience.While the code is complex, we have added comprehensive comments throughout to aid understanding of the model's framework and logic.

We believe these actions significantly improve the transparency and replicability of our study. We are grateful for the reviewer's insights and are ready to provide further clarification if needed.

3. Response to Reviewer #2

We sincerely thank the reviewer for their insightful comments regarding conceptual clarity and interpretive coherence. To address this core concern, we have added a synthesising overview paragraph at the end of the Introduction section. This addition explicitly integrates the key components of our theoretical framework and analytical approach, thereby strengthening the logical flow and narrative consistency throughout the manuscript.

We appreciate the reviewer's guidance in enhancing the manuscript's readability and theoretical rigour. We are happy to provide further clarification on any specific points if required.

---

## [Decision Letter · Decision Letter 1]

15 Feb 2026

PONE-D-25-58355R1The orchestrating role of platform enterprises in digital inclusion: A network game perspective from industrial clustersPLOS One

Dear Dr. Wang,

Thank you for submitting your manuscript to PLOS ONE. After careful consideration, we feel that it has merit but does not fully meet PLOS ONE’s publication criteria as it currently stands.

While one of the reviewers considers that the manuscript has satisfactorily addressed the previous concerns and is now technically sound, another reviewer raises substantial methodological and conceptual reservations that cannot be overlooked.

However, some concerns remain. These issues directly affect the robustness and credibility of the study. Therefore, the methodological section must be substantially strengthened. At this stage, the manuscript cannot proceed toward acceptance until these methodological concerns are convincingly addressed.

We therefore invite you to submit a thoroughly revised version of the manuscript, together with a detailed point-by-point response explaining how these issues have been resolved.

We look forward to receiving your revised manuscript.

Kind regards,

Juan E. Trinidad-Segovia, PhD

Section Editor

PLOS One

Journal Requirements:

Reviewers' comments:

Reviewer's Responses to Questions

**Comments to the Author**

1. If the authors have adequately addressed your comments raised in a previous round of review and you feel that this manuscript is now acceptable for publication, you may indicate that here to bypass the “Comments to the Author” section, enter your conflict of interest statement in the “Confidential to Editor” section, and submit your "Accept" recommendation.

Reviewer #1: (No Response)

Reviewer #2: All comments have been addressed

2. Is the manuscript technically sound, and do the data support the conclusions?

Reviewer #1: Partly

Reviewer #2: Yes

3. Has the statistical analysis been performed appropriately and rigorously? 

Reviewer #1: Yes

Reviewer #2: Yes

4. Have the authors made all data underlying the findings in their manuscript fully available?

Reviewer #1: No

Reviewer #2: Yes

5. Is the manuscript presented in an intelligible fashion and written in standard English?

Reviewer #1: Yes

Reviewer #2: Yes

6. Review Comments to the Author

Reviewer #1: Dear authors! After reviewing the revised version of the manuscript, I consider it necessary to note that the changes made by the authors are primarily formal and editorial in nature and do not address the key methodological and conceptual concerns raised during the first round of review. The status of the model and data used remains insufficiently defined, and the interpretation of the results goes beyond the limits of valid conclusions acceptable for theoretical simulation analysis. The methodological section, despite being partially expanded, does not provide the necessary transparency in the description of the parameters, scenarios, and modeling assumptions, which limits the reproducibility and analytical rigor of the study.

Reviewer #2: Dear Authors,

Thank you for submitting the revised version of your manuscript and for your careful engagement with the review comments.

The revision has satisfactorily addressed the substantive issues raised in the previous round, and the manuscript now presents a clear and technically sound contribution. I do not have further concerns regarding the methodology, data, or validity of the results.

At this stage, my remaining comments are minor and purely editorial in nature. I would suggest a final round of polishing focused on (i) ensuring consistent use of key concepts throughout the manuscript, (ii) improving clarity and flow in some parts of the discussion, and (iii) checking overall consistency in presentation (notation, figures, and references).

With these minor points addressed, the manuscript should be ready for publication.

Kind regards,

Reviewer

7. PLOS authors have the option to publish the peer review history of their article (what does this mean? ). If published, this will include your full peer review and any attached files.). If published, this will include your full peer review and any attached files.

**Do you want your identity to be public for this peer review?** For information about this choice, including consent withdrawal, please see our For information about this choice, including consent withdrawal, please see our Privacy Policy ..

Reviewer #1: No

Reviewer #2: **Yes:** Rudy Fernandez-EscobedoRudy Fernandez-Escobedo

---

## [Author Response · Author response to Decision Letter 2]

19 Feb 2026

Response to Reviewers

Manuscript ID: PONE-D-25-58355R1

Title: The orchestrating role of platform enterprises in digital inclusion: A network game perspective from industrial clusters

Dear Editor and Reviewers,

We would like to thank the editor and the reviewers for their thoughtful and constructive comments on our revised manuscript. We are pleased that Reviewer #2 acknowledges the manuscript as "technically sound" and suitable for publication following minor revisions, and we appreciate Reviewer #1's continued engagement in improving the methodological clarity and transparency of our work.

We have carefully addressed all remaining concerns raised by both reviewers. Below, we provide a point-by-point response outlining the revisions made.

Response to Reviewer #1

We thank Reviewer #1 for emphasising the importance of methodological transparency and conceptual clarity. In response, we have made the following revisions:

Clarification of the simulation-based nature of the study: In both the Abstract and Introduction, we have explicitly stated that this is a theoretical simulation study and that the data used are synthetically generated through Monte Carlo simulations. We have also included a description of the "Monte Carlo steps" to improve procedural transparency.

Enhancement of result robustness: To demonstrate the stability of our findings, we have added confidence intervals to all figures. Additionally, we have uploaded the raw data in .xlsx format to the OSF public repository and the PLOS ONE submission system to facilitate full reproducibility.

Improvement of methodological transparency: We have included a new Fig 1. Flowchart of the game procedure to visually illustrate the simulation process. We have also corrected several parameter inconsistencies and provided a detailed parameter table in the Appendix, specifying the rationale and literature basis for each parameter choice. Confirmed that all data and code are accessible via the provided OSF link.

https://osf.io/axtf9/overview?view_only=f39e90a6a71e40cd811aa5469a4aa9f6

We hope these revisions adequately address your concerns and enhance the clarity and rigour of the manuscript.

Response to Reviewer #2

We are grateful for your positive assessment and your recognition that the manuscript is now technically sound. We also appreciate your helpful suggestions for final polishing. In response, we have made the following editorial improvements:

Conceptual consistency: We have systematically reviewed the manuscript to ensure consistent use of key terminology throughout, including terms such as "platform enterprise," "investment strategy," "free-rider," and "connection ratio (CR)."

Language and readability: We have carefully proofread the manuscript and refined the language in several sections, particularly in the discussion, to improve clarity and fluency.

Consistency in presentation: We have revised all figures for visual consistency, standardised notation across the manuscript, and provided a comprehensive parameter table in the Appendix to complement the methodological description.

We believe these revisions have further strengthened the manuscript and we thank you once again for your constructive feedback.

We hope that the revised manuscript now meets the standards for publication in PLOS ONE and we look forward to your final decision.

---

## [Decision Letter · Decision Letter 2]

8 Mar 2026

The orchestrating role of platform enterprises in digital inclusion: A network game perspective from industrial clusters

PONE-D-25-58355R2

Dear Dr. Wang,

We’re pleased to inform you that your manuscript has been judged scientifically suitable for publication and will be formally accepted for publication once it meets all outstanding technical requirements.

Kind regards,

Juan E. Trinidad-Segovia, PhD

Section Editor

PLOS One

Additional Editor Comments (optional):

Reviewers' comments:

Reviewer's Responses to Questions

**Comments to the Author**

1. If the authors have adequately addressed your comments raised in a previous round of review and you feel that this manuscript is now acceptable for publication, you may indicate that here to bypass the “Comments to the Author” section, enter your conflict of interest statement in the “Confidential to Editor” section, and submit your "Accept" recommendation.

Reviewer #1: (No Response)

2. Is the manuscript technically sound, and do the data support the conclusions?

Reviewer #1: (No Response)

3. Has the statistical analysis been performed appropriately and rigorously? 

Reviewer #1: Yes

4. Have the authors made all data underlying the findings in their manuscript fully available?

Reviewer #1: Yes

5. Is the manuscript presented in an intelligible fashion and written in standard English?

Reviewer #1: Yes

6. Review Comments to the Author

Reviewer #1: Dear authors! After making revisions based on the reviewers' comments, the article has become more structured and well-developed.

7. PLOS authors have the option to publish the peer review history of their article (what does this mean? ). If published, this will include your full peer review and any attached files.). If published, this will include your full peer review and any attached files.

**Do you want your identity to be public for this peer review?** For information about this choice, including consent withdrawal, please see our For information about this choice, including consent withdrawal, please see our Privacy Policy ..

Reviewer #1: No

---

## [Editor Report · Acceptance letter]

PONE-D-25-58355R2

PLOS One

Dear Dr. Wang,

I'm pleased to inform you that your manuscript has been deemed suitable for publication in PLOS One. Congratulations! Your manuscript is now being handed over to our production team.

Kind regards,

on behalf of

Dr. Juan E. Trinidad-Segovia

Section Editor

PLOS One